# On-Device Deployment of Cerviray AI: Optimization via Knowledge Distillation and Quantization for Mobile Clinical Environments

**Byeongin Moon**[*1]                                    BLUEIN@AIDOT.AI

**Jaeyun Song**[*2]                                    SJYUNI105@GMAIL.COM

**Dongha Lee**[*1]                                    DHLEE@AIDOT.AI

**Seongmin Kim**[2]                                    NAIAD515@GMAIL.COM

**Donghoon Suh**[3]                                    SDHWCJ@NAVER.COM

**Hansol Choi**[†1]                                    SOLCHOI@AIDOT.AI

**Jaehoon Jeong**[†1]                                    JMAN@AIDOT.AI

[1] *AIDOT Inc., Seoul, South Korea*

[2] *Department of Obstetrics and Gynecology, Korea University Anam Hospital, Seoul, South Korea*

[3] *Department of Obstetrics and Gynecology, Seoul National University Bundang Hospital, Seongnam, South Korea*

## Abstract

Artificial intelligence has significantly advanced the diagnostic accuracy of Visual Inspection with Acetic acid (VIA) for cervical cancer screening. However, to overcome the GPU dependency of deep learning models and ensure their applicability in point-of-care settings, we present an on-device version of Cerviray AI. By employing Knowledge Distillation (ViT-Base to ViT-Tiny) and INT8 Post-Training Quantization, we successfully migrated the system from an RTX 4060 GPU to a Samsung Galaxy Tab S7 CPU. The optimized model achieves a clinical-grade accuracy of 97.61% (only a 0.24% drop) and an inference speed of 3.4s per image. This work demonstrates the potential of edge-AI in democratizing high-fidelity cancer screening for resource-limited, decentralized settings.

**Keywords:** Cervical Cancer, On-device AI, Knowledge Distillation, Model Quantization, Vision Transformer.

## 1. Introduction

Cervical Intraepithelial Neoplasia (CIN) grading is vital for determining treatment strategies. Cerviray AI automates this process, with efficacy validated through clinical trials (Kim et al., 2022, 2023) and its real-world effectiveness demonstrated in a recent prospective field study (Harsono et al., 2025). However, reliance on server-grade GPUs limits its clinical utility in point-of-care (POC) settings, particularly in low- and middle-income countries (LMICs) with unstable connectivity. Additionally, cloud-based processing raises significant data privacy concerns regarding sensitive patient images. To ensure robustness, we optimize the model on the Vision Transformer (ViT) (Dosovitskiy et al., 2021)—via Knowledge Distillation (KD) (Hinton et al., 2015) and Post-Training Quantization (PTQ) (Jacob et al., 2017) to run natively on mobile CPUs, enabling high-fidelity screening without cloud infrastructure.

---

[*] Contributed equally

[†] Corresponding author

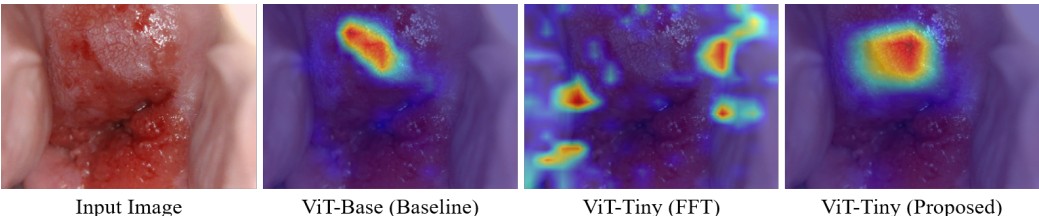

| Input Image | ViT-Base (Baseline) | ViT-Tiny (FFT) | ViT-Tiny (Proposed) |

Figure 1: Comparison of attention heatmaps. The proposed model shows high spatial alignment with the teacher (ViT-Base) mosaic pattern focus (Cosine Similarity: 0.91), whereas the fully fine-tuned(FFT) model exhibits scattered attention (0.68).

## 2. Dataset and Methodology

We utilized a multicenter dataset of 31,529 cervicography images (IRB approved). The data was split into Training (68%), Validation (16%), and Testing (16%) sets. Each image was labeled using the histopathological gold standard, verified by expert colposcopists.

### 2.1. Optimization Pipeline

The optimization focused on reducing architectural complexity and bit-precision while preserving spatial features necessary for medical imaging. The overall workflow of our two-stage optimization is illustrated in Figure 2.

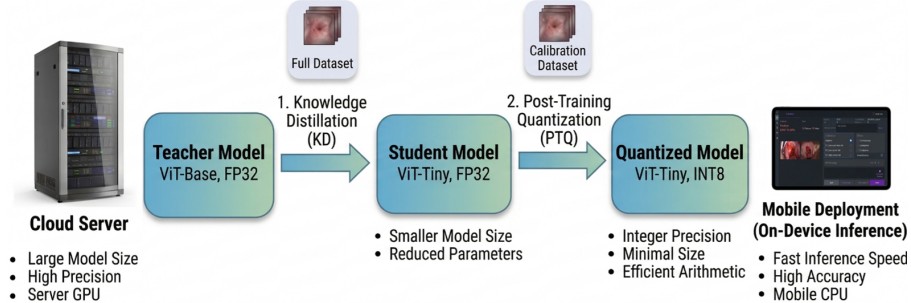

Figure 2: Optimization pipeline

**Knowledge Distillation (KD)** We distilled the diagnostic expertise of a ViT-Base (Teacher, $\mathcal{T}$) into a ViT-Tiny (Student, $\mathcal{S}$) following the DeiT approach (Touvron et al., 2021). The student was trained using a multi-objective loss function:

$$L_{total} = \alpha L_{CE}(y, \sigma(z_{\mathcal{S}})) + (1 - \alpha)\tau^2 KL(\sigma(z_{\mathcal{T}}/\tau), \sigma(z_{\mathcal{S}}/\tau))$$

where $L_{CE}$ is Cross-Entropy loss and $KL$ is Kullback-Leibler divergence. This allows the Student to capture the complex feature representations necessary for subtle CIN grading.

**Post-Training Quantization (PTQ)** Following KD, the model was converted from FP32 to INT8 using Static PTQ (Gholami et al., 2021). We used a calibration set of 400 images balanced across the four CIN grades. The quantization mapping is defined as:

$$Q(x) = \text{clip}(\lfloor x/S \rceil + Z, q_{min}, q_{max})$$

where $S$ is the scaling factor and $Z$ is the zero-point. This reduces memory bandwidth and computational load on the ARM-based mobile CPU.

## 3. Experiments and Results

### 3.1. Implementation and Hardware

Models were trained on an NVIDIA RTX 4060 GPU (8GB VRAM). For clinical validation, we deployed the optimized model on a Samsung Galaxy Tab S7 (Qualcomm Snapdragon 865+, 8GB RAM), reflecting the actual *Cerviray Expert* hardware used in POC environments.

Table 1: Comprehensive performance comparison of model optimization strategies. Values represent mean $\pm$ SD ($n = 5$). The proposed ViT-Tiny (KD+Static PTQ) maintains clinical-grade sensitivity with no statistically significant performance degradation compared to the teacher model.

| Metric | ViT-Base (Teacher) | ViT-Base (INT8 Dyn) | ViT-Tiny (FFT) | ViT-Tiny (KD) | ViT-Tiny (KD+Dyn) | ViT-Tiny (Proposed) |
|---|---|---|---|---|---|---|
| Input Size | 512 | 512 | 224 | 224 | 224 | **224** |
| Training | FFT | FFT | FFT | KD | KD | **KD** |
| PTQ | — | INT8 Dynamic | — | — | INT8 Dynamic | **INT8 Static** |
| Model Size | 344 MB | 88 MB | 23 MB | 23 MB | 6.5 MB | **6.5 MB** |
| Latency (Server)[†] | $1.5 \pm 0.1$s | — | $1.1 \pm 0.1$s | $1.1 \pm 0.1$s | — | — |
| Latency (Mobile)[‡] | — | $8.2 \pm 0.4$s | — | — | $3.8 \pm 0.2$s | **3.4 $\pm$ 0.2s** |
| Accuracy (%) | $97.85 \pm 0.08$ | $97.79 \pm 0.10$ | $86.94 \pm 0.52$ | $97.61 \pm 0.11$ | $96.53 \pm 0.14$ | **97.48 $\pm$ 0.12** |
| AUC | $0.986 \pm 0.002$ | $0.985 \pm 0.003$ | $0.965 \pm 0.007$ | $0.983 \pm 0.002$ | $0.976 \pm 0.004$ | **0.981 $\pm$ 0.003** |

FFT: Full Fine-Tuning, KD: Knowledge Distillation.

### 3.2. Performance Analysis

As summarized in Table 1, our proposed ViT-Tiny (KD+Static PTQ) model achieves a significant reduction in model size (344 MB to 6.5 MB) compared to the teacher model. While the transition to a mobile CPU environment increases latency to 3.4s, it remains well within the clinical threshold for real-time diagnostics. Crucially, the integration of KD effectively bridges the accuracy gap, recovering the student's baseline performance from 86.94% to 97.4%, with only a negligible 0.24% drop from the teacher model. This quantitative recovery is qualitatively reinforced by the attention heatmaps in Figure 1. This confirms that the combined KD and Static PTQ approach successfully preserves diagnostic integrity while enabling efficient on-device deployment.

## 4. Conclusion

We successfully engineered an on-device version of Cerviray AI by optimizing its core architecture for mobile CPUs. By leveraging advanced model compression, we bridged the gap between server performance and mobile accessibility, enabling democratized cancer screening in resource-limited settings.

## Acknowledgments

This work was conducted by AIDOT Inc. The authors are grateful to Korea University Anam Hospital and Seoul National University Bundang Hospital for their essential support in providing the clinical datasets, which were accessed and used under the approval of their respective Institutional Review Boards (IRBs).

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
