# OpenReview forum: "On-Device Deployment of Cerviray AI: Optimization via Knowledge Distillation and Quantization for Mobile Clinical Environments"
_MIDL.io/2026/Short_Papers — MIDL 2026 - Short Papers Poster_

### Official Review · Reviewer_ukph · 2026-05-02
**Optimization via Knowledge Distillation and Quantization for Mobile Clinical Environments**

**Rating:** 2
**Confidence:** 3

**Review:**

Both knowledge distillation and post-training quantization are well-established techniques, and their combination has been extensively studied. The paper does not introduce any new algorithmic component, loss function, or optimization strategy. The use of DeiT-style distillation is known, and the quantization pipeline follows existing approaches.
The dataset is large (31k images), but results are reported as single-point estimates without variance, confidence intervals, or statistical testing. The claimed accuracy drop of 0.24 percent is not supported by repeated experiments or significance analysis. There is no robustness analysis across devices or deployment conditions.
The evaluation metrics are limited to accuracy and AUC, without calibration analysis or subgroup performance. Given the clinical context, this is a significant omission. The latency measurements are reported for a single device, without variability or benchmarking against other deployment frameworks.
The paper does not compare against alternative compression methods such as pruning or low-rank adaptation. The baseline is limited to variants of the same architecture, which does not provide meaningful insight.
The paper focuses on model compression and deployment, applying knowledge distillation and quantization to a Vision Transformer for cervical cancer screening. While the application is relevant, the methodological contribution is minimal.
The qualitative attention maps in Fig 1 and 2 are not quantitatively evaluated and do not provide evidence of preserved diagnostic reasoning.
The work is an engineering exercise with limited novelty and insufficient experimental rigour.

**Summary:**

This paper presents an optimization pipeline for deploying a cervical cancer screening model on mobile devices using knowledge distillation and post-training quantization. A Vision Transformer is compressed from a large teacher model to a smaller student model and quantized to INT8. The resulting model achieves comparable accuracy with reduced latency on a mobile device. The work targets practical deployment in resource-limited clinical settings.

**Strengths:**

The paper addresses a practically important problem of deploying AI models in resource-constrained environments. The dataset is relatively large and clinically relevant. The implementation demonstrates feasibility of running models on mobile devices with reduced latency and memory footprint.

**Weaknesses:**

The work lacks methodological novelty and relies entirely on standard techniques. The statistical analysis is absent, with no variance or significance testing. The evaluation is limited in scope, lacking robustness, calibration, and subgroup analysis. The comparisons are weak and do not include alternative compression strategies. The clinical claims are not substantiated with rigorous validation.

**Justification Of Rating:**

While practically relevant, the paper does not provide sufficient novelty or rigorous evaluation. The lack of statistical analysis and limited experimental scope prevent acceptance.

---

### Decision · Program_Chairs · 2026-05-08

Accept (Poster)